# Effect of Rearing Systems on Growth Performance, Lying/Standing Behavior, Morbidity, and Immunity Parameters of Pre-Weaned Dairy Calves in a Continental Zone in Winter

Wanying Zhao [1,2], Christopher Choi [3], Lin Ru [4], Zhengxiang Shi [1,2,*] and Hao Li [1,2]

[1] College of Water Resources and Civil Engineering, China Agricultural University, Beijing 100083, China
[2] Key Laboratory of Agricultural Engineering in Structure and Environment, Ministry of Agriculture and Rural Affairs, Beijing 100083, China
[3] Department of Biological Systems Engineering, University of Wisconsin-Madison, 460 Henry Mall, Madison, WI 53706, USA
[4] College of Engineering, Heilongjiang Bayi Agricultural University, Daqing 163319, China
[*] Correspondence: shizhx@cau.edu.cn

**Abstract:** On dairy farms, calves are typically raised inside barns (either in individual or group pens), or they are raised in outdoor hutches. To evaluate the effect of all three of these commonly used rearing practices on calves, an experiment was conducted. A group of 58 Holstein dairy female healthy calves (3 days of age) was randomly divided into three subgroups (outdoor hutches, individual-housed, and group-housed in a barn). The body weight, lying bouts, lying time, and immunity parameters of each calf were monitored, and the ambient temperature and relative humidity were measured. The average temperatures outside and in the barn and hutches were $-16.67\ ^\circ$C, $-15.26\ ^\circ$C, and $-15.59\ ^\circ$C, respectively, from 22 November 2020 to 27 January 2021. All calves suffered from cold stress. Group-housed calves weighed significantly less than the other calves at the ages of 1 month and 2 month ($p < 0.05$). The lying time of the calves housed in individual pens and group pens was longer ($p < 0.05$) than that of the calves housed in hutches. The morbidity attributable to bovine respiratory disease was significantly lower among the calves housed in hutches than it was among the calves housed either individually or in group pens inside the barn ($p < 0.05$). No significant differences in the concentrations of TNF-$\alpha$, IL-1$\beta$, and IL-6 were found between the three groups ($p > 0.05$). On the basis of these findings, we were able to conclude that calves housed in outdoor hutches were at a lower risk of developing a disease than were calves housed in barns without heating in winter. To optimize the management process, heating should be added to hutch systems. Moreover, more rigorous disease and environmental control management strategies should be applied when raising calves inside barns.

**Keywords:** hutch; barn; average daily weight gain; lying time; morbidity rate



## 1. Introduction

The pre-weaning period (the first 8 weeks of a calf's life) is a critical time, during which daily weight gain determines the rearing costs and initial milk yields of female calves in their future. Furthermore, in general, a pre-weaned calf raised under poor management practices would be more likely to suffer bovine respiratory disease (BRD) and diarrhea and to succumb to severe morbidity or even death, any of which can cause significant economic loss [1,2]. Calf respiratory disease, for example, is estimated to cost the UK cattle industry GBP 80 million annually (between GBP 30 for a mild case to GBP 500 when an animal dies) [3]. Such negative impacts should call for strategies aimed at reducing the severity of the effects that these diseases can impose; as it is well known that housing and management practices can influence the risk levels associated with the

pre-weaned period [4,5], developing a strategy for choosing an appropriate housing system and optimizing it in a targeted manner would seem especially relevant.

As one recent study concluded, the ambient temperature was found to be important, with both high and low temperatures being associated with increased mortality among pre-weaned calves of 15–55 d, while associations in weaned calves (56 d—1 year) were only observed for low temperatures [6]. Cold stress occurs when the ambient temperature falls below the calf's lower critical temperature limit, which varies with age. Chronic cold stressors can affect animals by altering both antibody-mediated and cell-mediated immunity [7]. The immunocompetence of a calf before its first mixing can greatly influence whether it will succumb to postweaning morbidity and mortality.

There is a dearth of information that describes how housing systems may influence the immune responsiveness of bovine calves and, ultimately, infectious disease resistance in winter. Calves raised together inside a traditional barn, on the other hand, are significantly more likely to spread germs than calves housed in individual hutches or pens, and thus, infect other members of the group [8,9]. Current research has, however, found that dairy producers will choose to group their pre-weaned calves to reduce the amount of labor needed to feed the animals, and also because group feeding socializes calves earlier in life and provides growth rates equivalent to those achieved by individual housing [10]. Average daily weight gain (ADWG) is considered an appropriate metric for evaluating growth and health during the preweaning period [11], and indeed, at least one earlier study found evidence suggesting that housing calves in a group promote both growth and social behavior because calves housed in groups tend to ingest more starter intake and, in turn, register a higher ADWG (16.0%) than individually housed calves [12]. Social learning may also encourage a calf to eat more solid feed, which, in turn, promotes weight gain [10]. Although producers and veterinarians continue to express concern that group housing may encourage more health issues, a study by Kung et al. [13] found that the number of treatments involving medications such as antibiotics was the same whether calves were raised in group housing or individual housing. Producers using calf hutches reported lower death rates among pre-weaned calves than those of producers using other housing types [14]. However, a study concluded that housing calves indoors most likely reduces mortality rates because doing so protects the animals from direct sunlight during hot weather [15] and also helps them to maintain their core body temperature during cold weather [16]. Since few studies before this one had sought to quantify and compare the health effects associated with the three different rearing systems in winter, and, in particular, how those systems might compare if applied at the same dairy farm, a multitude of questions remained unanswered regarding how a rearing system may affect the health of a calf.

Consequently, this study was designed to systematically evaluate all three of the housing options that are currently used to raise pre-weaned dairy calves in winter. The growth performance, health, behavior, and immunity parameters specific to rearing systems were considered.

## 2. Materials and Methods

### 2.1. Experimental Design

A trial was conducted at a Holstein dairy farm (47°55′ N, 126°23′ E) in the Heilongjiang Province of China. The weather in this region, which is characterized as "zone D—continental zone" by the Köppen climate classification system, is similar to the weather across much of Europe, as well as the American Midwest and Northeast.

All experimental producers were approved by the Institutional Animal Care and Committee (LACUC approval no. AW90901202-5-1) of China Agricultural University. The trial lasted from 22 November 2020 to 27 January 2021. Fifty-eight healthy female calves (average 3 days of age) of similar weight (36–40 kg) were selected and were randomly assigned to either a hutch outdoor (n = 25) (Figure 1A), an individual pen inside the barn (n = 9) (Figure 1B) or a group pen inside the barn (n = 24, 8 calves per pen) (Figure 1C).

Hutches were constructed of polyethylene and were 1.22 m wide by 2.20 m long at the base and 1.36 m in height. The individual pen was 3.60 × 1.20 × 1.07 m and partitioned by wire mesh on each side, and the pens were immediately adjacent to each other, which allowed for calf contact. The group pen was 12.0 × 6.0 × 1.20 m. The calf barn (180.0 × 42.0 × 7.20 m) contained 4 rows of pens and was ventilated using a natural ventilation system. Vents were located on both sidewalls, they were 2.4 m high, and they ran the entire length of the barn. The northside vent was only open to approximately 0.2–0.5 m in winter, while the southside vent was open to approximately 1.0–1.5 m on sunny days. The doors were kept closed in winter, except to admit the passage of necessary airflow to avoid an odorous and humid microenvironment. A schematic is shown in Figure 1D.

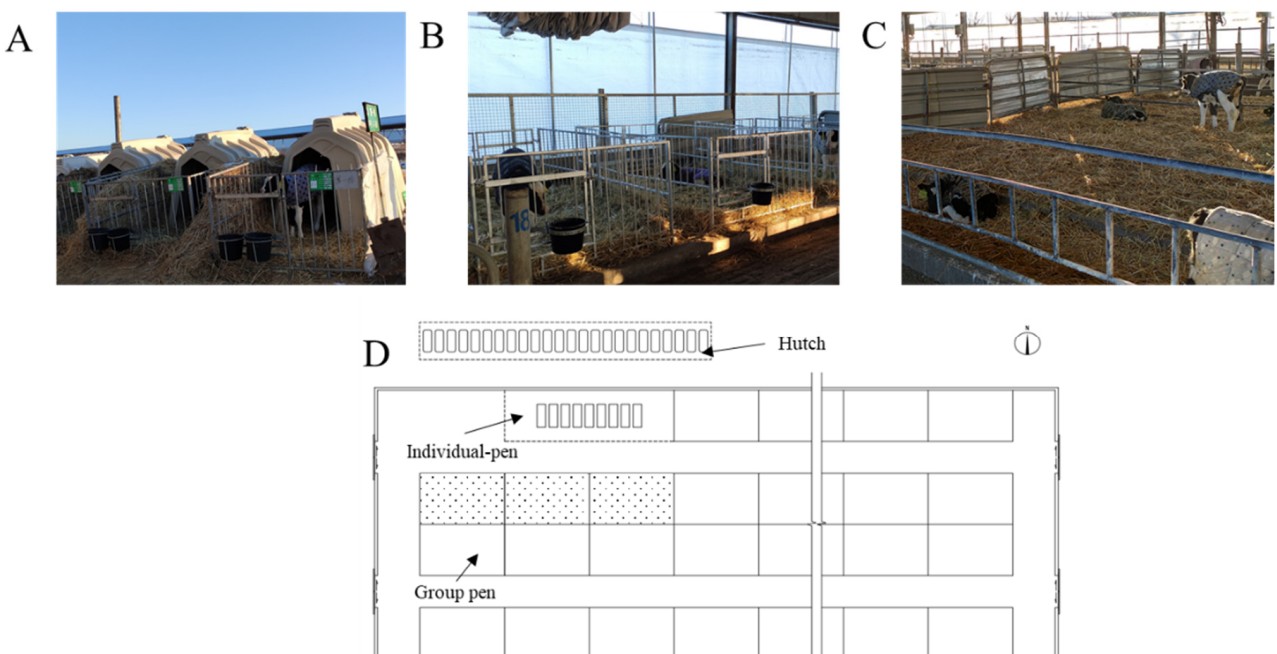

**Figure 1.** The photos showing (**A**) outdoor hutches, (**B**) individual pens, and (**C**) group pens inside the barn, and the schematic for the (**D**) calf barn.

The calves were separated from their dams within 2 h after birth and fed 4 L of frozen colostrum, which was manually bottle-fed to the calves. The calves were managed the same across the rearing systems and fed two times a day, i.e., at 0700 h and 1300 h. The milk fed to the calves was provided according to this schedule: from d 1 to d 10, 4 L per day; from d 11 to d 30, 6 L per day; from d 30 to d 52, 8 L per day; from d 53 to d 55, 6 L per day; from d 56 to d 57, 4 L per day; from d 58 to d 60, 2 L per day. The calves were weaned at d 60 by feeding them milk only in the morning on d 58–60. On d 4, the calves were provided ad libitum access to the starter diet (≤14% moisture, ≥20% crude protein, ≤12% crude fiber, ≤9% crude ash, 0.7–0.8% calcium, ≥0.4% total phosphorus, 0.3–1.5% sodium chloride, ≥0.7% lysine). Those calves reared in group pens were provided free-choice water, and those calves raised in individual pens and hutches were offered water from a plastic bucket (9.5 L) filled twice a day. All the calves wore calf jackets made at the farm to protect them from the cold weather.

### 2.2. Measurements

The calves' total serum protein concentrations were tested using an optical refractometer on d 3 and were found to be over 5.2 g/L. A total serum protein concentration > 5.2 g/L is often considered indicative of an adequate passive transfer of immunity in healthy, well-hydrated calves [17]. A digital scale was used to measure each calf's body weight (BW) within hours after birth, then again at d 30, and again at weaning. The pre-weaning

weight gain was calculated by subtracting the birth weight from the 2-month weight. The ADWG was calculated by taking the weight gain during the preweaning period divided by the number of days between the birth weighing and the 2-month weighing. Air temperature and humidity were recorded hourly using a data logger (HOBO U23 Pro v2; Onset Computer Co., Ltd., Bourne, MA, USA; temperature accuracy of $\pm 0.25$ °C, range: $-40$ to 70 °C; relative humidity accuracy of $\pm 2.5\%$ from 10% to 100%). Loggers were mounted 1.0 m from the floor at the center of each group pen and each individual pen, on the inside wall of an occupied calf hutch at 0.5 m from the ground, and in continuous shade outside at 1.5 m from the ground. Loggers in the hutches were encased in corrugated plastic to prevent contact with the calves.

### 2.3. The Definition and Diagnosis of Illness

A qualified veterinarian determined whether each calf had diarrhea (fecal score 1 to 3; Table 1) [18]. If a calf refused milk or registered an individual fecal score $\geq 2$, it was diagnosed as diarrhea by the veterinarian and treated accordingly. With respiratory disease being the second most common factor related to calf mortality [19], each calf's eyes and cough matter were examined and scored by a trained observer. The health status was assessed once daily at 1300 h over the first 21 d. Each sick calf was treated by a veterinarian and the calf was not excluded from the trials.

**Table 1.** Calf health factor scoring system.

| Health Factor | Scoring System | | | |
|---|---|---|---|---|
| | 0 | 1 | 2 | 3 |
| Fecal score | Normal | Semi-formed, pasty | Loose, but stays on top of bedding | Watery, sifts through bedding |
| Eye score | Normal | Small amount of ocular discharge | Moderate amount of bilateral discharge | Heavy ocular discharge |
| Cough | Normal | Induced single cough | Induced repeated coughs or occasional spontaneous cough | Repeated spontaneous coughs |

### 2.4. Calf Lying and Standing Behaviors and Blood Sample Collections and Measurements

The lying and standing behavior of the calves were recorded using an electronic data logger (HOBO Pendant G Acceleration Data Logger, Onset Computer Co., Ltd., Pocasset, MA, USA) that was attached to the medial side of the front leg of each calf in a position such that the *x*-axis was parallel to the ground, the *y*-axis was perpendicular to the ground and pointing upward, and the *z*-axis was parallel to the ground and pointing away from the sagittal plane. The data loggers recorded the g-force on the *x*-, *y*-, and *z*-axes at 1 min intervals for 32 d, where the entire period was preprogrammed to begin on 5 December 2020. The g-force readings were converted into degrees of tilt, and the degree of vertical tilt (*y*-axis) was used to determine the lying position of the animal, such that readings $\geq 60°$ indicated that the cow was standing, whereas readings $<60°$ indicated that the cow was lying down [20]. Lying bouts of $<2$ min was ignored because these readings were likely associated with leg movements at the time of recording [21].

To determine total concentrations of immunoglobulin A (IgA), immunoglobulin G (IgG), immunoglobulin M (IgM), tumor necrosis factor-$\alpha$ (TNF-$\alpha$), interleukin-1$\beta$ (IL-1$\beta$), and interleukin-6 (IL-6), blood samples were collected using a vacuum blood collector (without anticoagulant) in pens, which was conducted by a veterinarian on the farm. The samples were kept at room temperature until the blood clotted; then, the serum was separated via centrifugation at $4000\times g$ for 10 min, aliquoted, and stored frozen ($-20$ °C) until the samples could be assayed. The concentrations of IgA, IgG, IgM, TNF-$\alpha$, IL-1$\beta$, and IL-6 were detected using commercially available Enzyme-Linked Immunosorbent Assay (ELISA) kits (Beijing Kang Jia Hong Yuan Biological Technology Co., Ltd., Beijing, China).

The blood samples were collected three times at weeks 1, 3, and 6 (i.e., at 7, 21, and 42 days of age) and from 8:00 am to 10:00 pm.

### 2.5. Statistical Analysis

All analyses were performed using the software SPSS (SPSS Inc. Released 2008. SPSS Statistics for Windows, Version 17.0. SPSS Inc., Chicago, IL, USA). For all groups, the mean morbidities of BRD and diarrhea were determined as the mean morbidities % = (sum of observed calf illness days/sum of all experiment days) × 100%. The BW, environmental temperature and humidity, serum immunity parameters, and behaviors were analyzed with repeated-measures analysis of variance (ANOVA). Differences between groups were deemed statistically significant if the associated $p$-value $\leq 0.05$. The immunity parameters data were analyzed using linear mixed models. The data were analyzed according to the fixed effects of raising systems and the ages of calves. The model's governing equation was as follows:

$$Y_{ijk} = \mu + RS_i + WA_j + R_k + RS_i \times WA_j + \varepsilon_{ijk}$$

where $Y_{ijk}$—parameters investigated, $\mu$—model constant, $RS_i$—effect of rearing system ($i$ = 1 to 3), $WA_j$—effect of age (week) ($j$ = 1, 3, and 6), $R_k$—replicate, $RS_i \times WA_j$—effect produced by the interaction of raising system and age, and $\varepsilon_{ijk}$—the residual error term.

## 3. Results

### 3.1. Ambient Temperature and Relative Humidity

In the trial, the daily average ambient temperature in the hutch ($-15.59\ °C$) and the barn ($-15.26\ °C$) were higher ($p < 0.05$) than outside ($-16.67\ °C$; Table 2), and the average daily relative humidity in the hutch (85.56%) was higher ($p < 0.05$) than outside (71.01%) or inside the barn (71.01%; Table 2).

**Table 2.** Ambient temperature and relative humidity outside and inside the barn and hutches during the trial.

| Elements | Mean ± SEM | | |
|---|---|---|---|
| | **Outside** | **Barn** | **Hutch** |
| Ambient temperature (°C) | $-16.67 \pm 0.19$ [b] | $-15.26 \pm 0.19$ [a] | $-15.59 \pm 0.38$ [a] |
| Relative Humidity (%) | $71.01 \pm 0.32$ [b] | $68.86 \pm 0.32$ [c] | $88.56 \pm 0.62$ [a] |

Note: [a–c] means within a row with different superscripts differ ($p < 0.05$).

### 3.2. Growth Performance and Behavior

In the trial, the rearing system did not affect the ADWG ($p > 0.05$), and group-housed calves weighed less than the other calves at 1 month and 2 months (Table 3). The group-housed calves had significantly lower DMIs ($p < 0.05$) than calves housed in individual pens in the barn and outdoor hutches. The DMIs of calves for each week are reported in Figure 2A. The DMIs increased with age, and at weeks 7–9, the DMIs of calves that were individually housed in the barn were significantly higher than the other calves. In the trial, those calves that were kept in individual pens spent an average of 18.71 h/d lying down, whereas the calves raised in group pens spent 18.06 h/d, and both of these groups spent more time ($p < 0.05$) lying down than did the calves kept in hutches (Table 3). Moreover, the number of lying bouts registered by the calves kept in hutches was higher ($p < 0.05$) than the number registered by the calves kept in individual pens (Table 3). The lying time per day of the calves for each week is reported in Figure 2B. At weeks 2, 3, and 5, the lying times registered by the calves kept in hutches were significantly shorter than the times registered by the calves housed inside the barn. At weeks 6–9, the calves spent approximately the same amount of time lying down, regardless of the rearing system.

**Table 3.** Performance of calves housed in the hutches, individual pens, or group pens in the barn in the trial.

| Item | Hutches | Barn | |
| --- | --- | --- | --- |
| | | Individual Pens | Group Pens |
| Calves, n | 24 | 9 | 24 |
| Initial serum protein (mg/dL) | 7.51 ± 0.13 | 7.44 ± 0.12 | 7.47 ± 0.07 |
| Initial BW (kg) | 36.96 ± 0.68 | 39.78 ± 1.05 | 36.43 ± 0.94 |
| 1-month BW (kg) | 54.87 ± 1.09 [a] | 50.50 ± 1.73 [ab] | 49.43 ± 1.36 [b] |
| 2-month BW (kg) | 81.96 ± 2.19 [a] | 81.78 ± 3.81 [a] | 73.59 ± 2.55 [b] |
| ADWG (kg/d) | 0.72 ± 0.03 | 0.70 ± 0.06 | 0.66 ± 0.03 |
| DMI (kg/d) | 0.33 ± 0.01 [a] | 0.32 ± 0.02 [a] | 0.25 ± 0.03 [b] |
| Lying time per day (h/d) | 16.28 ± 0.26 [b] | 18.71 ± 0.21 [a] | 18.06 ± 0.17 [a] |
| Lying bouts (no./d) | 13.84 ± 0.61 [a] | 11.98 ± 0.40 [b] | 13.18 ± 0.37 [ab] |

Note: BW—body weight; ADWG—average daily weight gain; DMI—dry matter intake. [a,b] means within a row with different superscripts differ ($p < 0.05$).

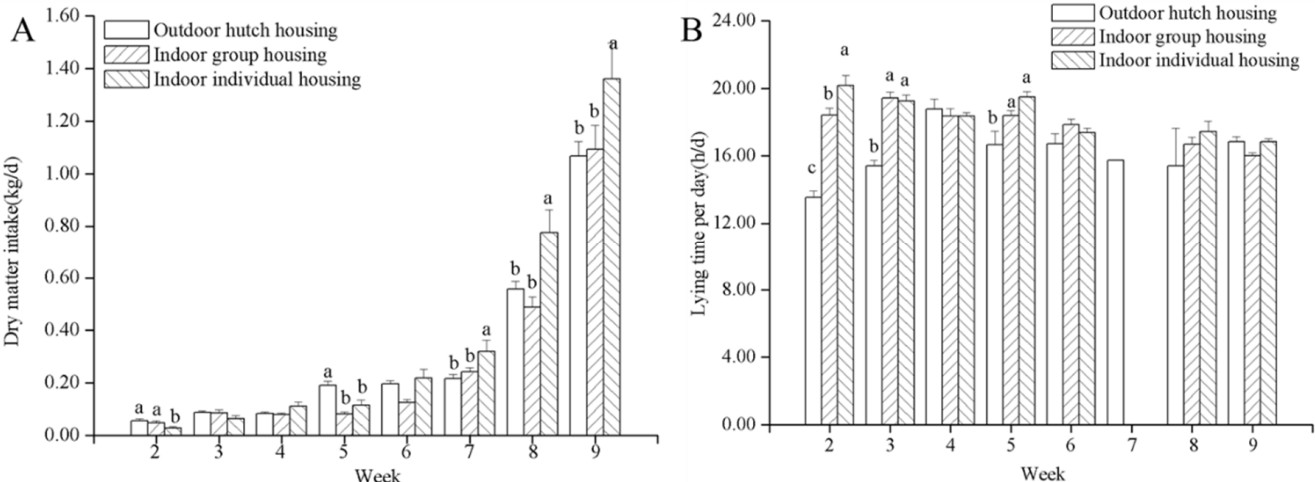

**Figure 2.** Relationship between (**A**) dry matter intake and (**B**) lying bouts of calves in the trial. Observations began at week 2 and ended at week 9 of age. [a–c] Means within a week for the rearing systems were significantly different at $p < 0.05$.

### 3.3. Health Status with Morbidity and Immunity Parameters

The rate of morbidity sustained by any group depended significantly on the type of rearing system used and the age of the calf. The mean morbidity rates related to BRD among calves housed in either a group pen, an individual pen inside the barn, or in hutches were 53.1%, 53.1%, and 23.7%, respectively ($p < 0.05$), and the mean morbidity rates related to diarrhea among calves housed in either a group pen, an individual pen inside the barn, or in hutches were 13.5%, 12.3%, and 10.1%, respectively ($p = 0.56$). The rates of morbidity among calves housed inside a barn were higher than the rates of those calves housed in hutches.

At weeks 1–2, the calves housed in hutches registered higher rates of morbidity (due to BRD) than did the calves housed inside the barn ($p < 0.01$). At weeks 3–5, the morbidity due to BRD among the calves housed inside the barn gradually increased, reaching a peak at week 5, and then the morbidity due to BRD gradually decreased, and at week 9 all the calves in all three rearing systems registered similar morbidities due to BRD (Figure 3A). At week 2, all the calves registered a higher rate of morbidity due to diarrhea than they had at any other testing point. The prevalence of diarrhea among the calves housed inside the barn ended at week 3, but the diarrhea rate among calves housed in hutches continued to hold steady until week 6 (Figure 3B).

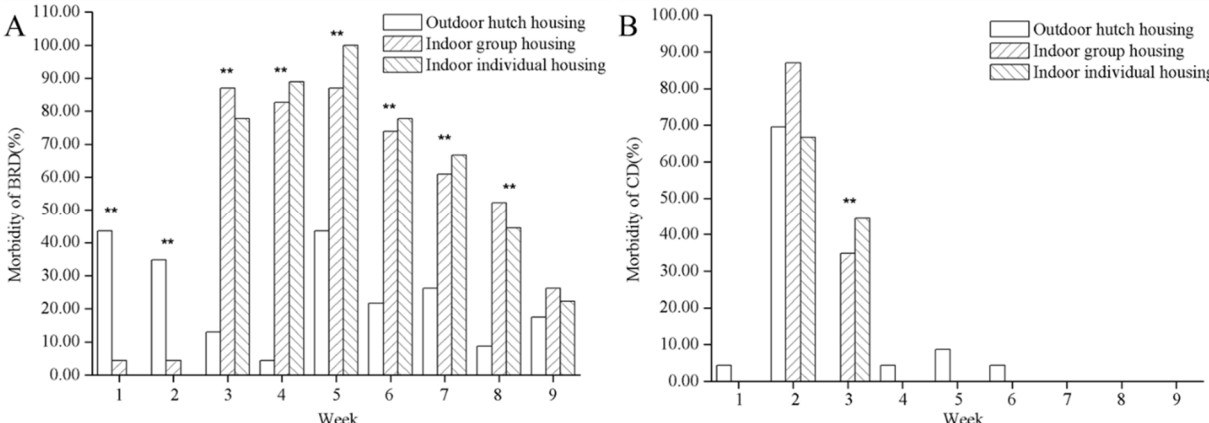

**Figure 3.** Morbidities as a result of (**A**) bovine respiratory disease (BRD) and (**B**) diarrhea among calves. ** Means within the same week of calves were significantly different at $p < 0.05$ between the three rearing systems.

As shown in Table 4, the combination of the rearing system and week significantly affected the concentration of IgM ($p < 0.05$). The concentration of IL-1β among the calves kept in individual pens inside the barn was significantly lower than it was among calves kept in the group pens and hutches ($p < 0.05$). At week 3, the concentration of IgG among calves kept in hutches was higher than it was among calves kept in the individual pens inside the barn ($p < 0.05$) (Figure 4). At week 6, the concentration of IgM among calves housed in hutches was higher than it was among calves housed in the barn ($p < 0.05$) (Figure 4). The type of rearing system used apparently did not affect the serum concentrations of IgA, TNF-α, IL-1β, and IL-6 among the calves of the same age.

**Table 4.** Immunity parameter values for the calves in the trial.

| Item | IgG (g/L) | IgA (g/L) | IgM (g/L) | TNF-α (pg/mL) | IL-1β (pg/mL) | IL-6 (pg/mL) |
|---|---|---|---|---|---|---|
| Rearing system | | | | | | |
| Outdoor hutch housing | 6.40 | 1.01 | 1.83 | 24.16 | 15.53 [b] | 61.57 |
| Indoor group housing | 6.33 | 1.02 | 1.10 | 22.79 | 15.49 [b] | 59.62 |
| Indoor individual housing | 6.02 | 1.08 | 0.99 | 22.33 | 17.14 [a] | 58.71 |
| SEM | 0.15 | 0.08 | 0.33 | 1.05 | 0.41 | 1.94 |
| Week | | | | | | |
| 1 | 5.97 [b] | 0.83 [b] | 0.74 [b] | 22.39 | 17.07 [a] | 61.97 |
| 3 | 6.12 [ab] | 0.98 [b] | 1.01 [b] | 23.61 | 16.10 [ab] | 60.63 |
| 6 | 6.66 [a] | 1.31 [a] | 2.17 [a] | 23.27 | 14.99 [b] | 57.31 |
| SEM | 0.19 | 0.27 | 0.07 | 1.03 | 0.46 | 1.86 |
| *p*-Value | | | | | | |
| Rearing system | 0.35 | 0.71 | 0.07 | 0.43 | 0.02 | 0.54 |
| Week | 0.03 | <0.01 | <0.01 | 0.69 | 0.01 | 0.20 |
| Rearing system × week | 0.07 | 0.27 | 0.03 | 0.17 | 0.81 | 0.89 |

Note: [a,b] values with different letters within the same item are significantly different ($p < 0.05$).

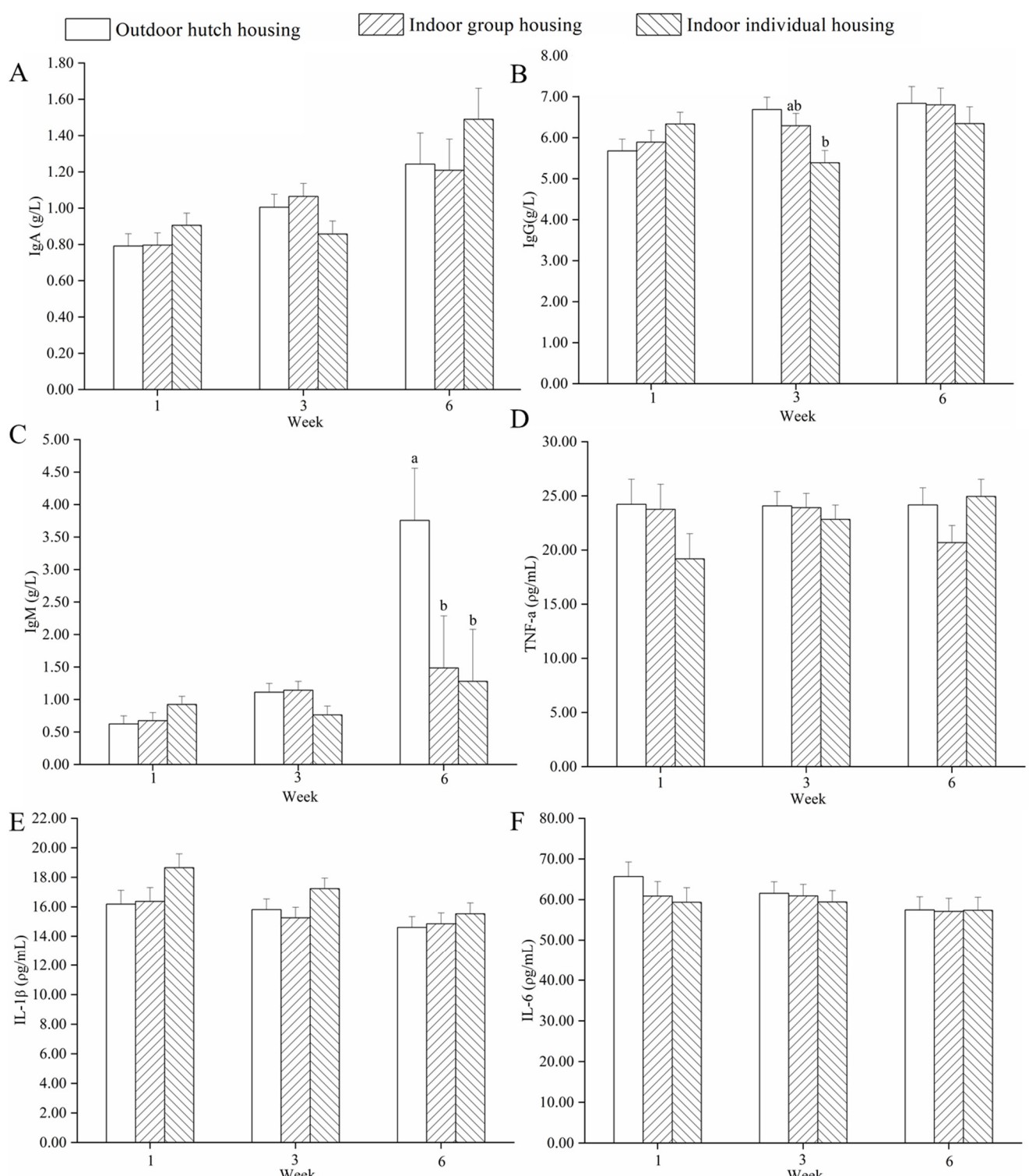

**Figure 4.** Concentrations of biochemical parameters of the calves in the different housing systems. (**A**) IgA, (**B**) IgG, (**C**) IgM, (**D**) TNF-α, (**E**) IL-1β, and (**F**) IL-6. Data represent mean ± SEM. [a,b] Values with different letters within the same week of calves were significantly different ($p < 0.05$) between the three rearing systems.

## 4. Discussion

This study systematically evaluated all three of the housing options (outdoor hutch housing, indoor group housing, and indoor individual housing) that are currently used to raise pre-weaned dairy calves in winter.

### 4.1. Ambient Temperature and Relative Humidity in the Dairy Farm

The thermoneutral zone associated with calves varies, ranging from 15 to 25 °C depending on age, weight, environmental conditions, and other stressors [22], and the physiological range of core body temperatures is between 38.1 °C and 39.3 °C [23]. The lower critical temperatures of calves will vary with age, with 13 °C being typical for a newborn calf and 8 °C being typical for calves up to 2 months of age [22]. Cold stress has been categorized as "mild" (0 °C to −6.7 °C), "moderate" (−7.2 °C to −13.9 °C), and "severe" (<−13.9 °C) under dry-winter cattle-coat conditions [24]. Apparently, in the trial, the ambient temperature was lower than the lower critical temperature of the calves; consequently, those calves suffered severe cold stress. The hutches provided a microclimate that was likely beneficial during cold, rainy, or windy conditions [25]. The calf hutch provided a relative improvement in the low-temperature environment outside the barn, which was related to the windproof and insulating nature of the calf hutch itself. The calf hutch provided a close approximation to the ambient temperature inside the barn.

### 4.2. Health Status, Growth Performance, Lying and Standing Behavior, and Immunity Parameters of the Tested Calves

In the trial, the morbidity rate among the calves housed inside the barn was higher than that of the calves kept in hutches, and this finding was similar to that of a previous study that showed that an indoor housing system may have adverse effects on a calf's health since multiple animals kept indoors share the same air, which can increase the risk of airborne disease transmission [26]. Some surveys suggested that the elevated relative humidity and the lower temperature that prevails in the stalls during the winter can increase the incidences of BRD and diarrhea [27]. Calves born in the winter had 2.6 times greater odds of being treated for BRD than those born in summer and 1.6 times greater than those born in the fall [5]. The mean disease incidence at a THI of ≤50 was significantly higher than that at a THI of ≥71 [28].

Cold stress and diseases can affect the weight and behavior of pre-weaned calves. Preweaning AWDG was shown to significantly impact the first lactation milk yield, with each 0.1 kg of AWDG being associated with an 85–113 kg increase in milk yield during first lactation [29]. Hyde et al. [30] showed that a 1 °C decrease in average ambient temperature during the calf's first month was associated with a 0.012 kg/d decrease in calf daily weight gain. Our data substantiated these findings. Previous research suggested that neonatal calf diarrhea is negatively associated with AWDG [5]. Calves that suffered at least one disease event gained 0.07 kg/d less than did calves that remained healthy [11]. This result may support findings from the current study showing that the morbidity rate among the calves housed inside a barn was higher than that of the calves kept in hutches, and the 2-month BW and AWDG of the calves kept in group pens were lower than those of other groups. Hyde et al. [30] showed that the environmental hygiene of group housing is consistently important for improving AWDG, potentially through reduced disease levels.

In dairy calves, the most sensitive indicator of BRD was lying time [31]. Similarly, increased lying times [32] and fewer lying bouts [33] were associated with BRD status when compared with healthy calves. Meanwhile, low temperature can increase the lying time of calves compared with calves in the thermoneutral zone [34]. Those results were similar to those found in this study. In the present study, the calves kept in hutches spent the least amount of time reclining and engaged in lying bouts more frequently than any other group, namely, those calves housed in a barn, and they had the least morbidity. The lying times and number of lying bouts registered by the calves kept individually did not differ significantly from the group-housed calves, and calves kept individually and in groups in the barn had the same morbidities of BRD.

The cost of antibiotics to treat the cold-environment (1.2–10.5 °C) calves was found to be higher at weeks 4 and 6 than the cost to treat calves raised in warm environments (average 15.5 °C), and treatment differences regarding respiratory scores and costs associated with antibiotic administrations were no different at week 7 [35]. Cold stress can cause

deviations from the normal immune response, and long-term exposure to stress might have immunosuppressive effects by making calves less resilient to diseases [36]. In the trial, the concentrations of TNF-α, IL-1β, and IL-6 among all the calves were found to be the same regardless of the rearing system (Figure 4), suggesting that the levels of immune activation among the three groups of calves were comparable.

In addition, the colostrum IgG concentration had a significant effect on the passive transfer status in calves [37]. Due to the limited content, colostrum IgG was not tested in this study, and subsequent studies can be conducted for related work. Appropriate colostrum and immunization management strategies should be used by producers and veterinarians in order to optimize passive transfer status and improve the growth performance of calves raised in production.

## 5. Conclusions

Each type of housing system offers certain advantages, but each type also has particular disadvantages. Calves kept in outdoor hutches, for example, are apparently at less risk of contracting diseases than are calves housed inside a barn without heating in winter; however, during cold winters, hutches cannot provide a warmer temperature than can a barn. Moreover, although calves housed inside a barn tend to suffer higher levels of morbidity than do calves housed in hutches (albeit the reasons for this phenomenon are complex and should therefore be studied in more depth), housing of any type reduces the effects of climate and, in fact, will likely provide calves with a beneficial microclimate. Nevertheless, we recommend that producers located in the severe cold zone avoid sheltering calves in hutches in winter unless a suitable heating system is employed. Overall, adequate heating systems should be developed for use in calf barns and hutches. That said, we should add that fully understanding the factors that would best describe the optimal housing system (one able to minimize the risks of pernicious diseases and temperatures under a wide variety of environmental and management conditions) will require more research. Regardless, whatever the housing system a dairy producer chooses, it should be one that can provide calves with a clean, dry environment and shelter from extreme solar radiation and heavy weather.

**Author Contributions:** W.Z.: conceptualization, methodology, investigation, data curation, formal analysis, writing—draft, writing—review and editing; C.C.: methodology and writing—review and editing; L.R.: investigation and data curation; Z.S. and H.L.: conceptualization, project administration, writing—review and editing, funding acquisition, and supervision. All authors read and agreed to the published version of the manuscript.

**Funding:** The study was supported by China Agriculture Research System of MOF and MARA.

**Institutional Review Board Statement:** The animal study protocol was approved by the Institutional Review Board of China Agricultural University (approval no. AW90901202-5-1 and 2021.09.21).

**Informed Consent Statement:** Not applicable.

**Data Availability Statement:** Not applicable.

**Acknowledgments:** The authors thank Huiyuan Guan and Xinyi Du of China Agricultural University for their assistance during the data collection.

**Conflicts of Interest:** There are no conflicts of interest for any of the authors.

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
