# Peer review of "Effect of Rearing Systems on Growth Performance, Lying/Standing Behavior, Morbidity, and Immunity Parameters of Pre-Weaned Dairy Calves in a Continental Zone in Winter"

_agriculture, doi:10.3390/agriculture12091496_

Round 1

Reviewer 1 Report

The manuscript is well written and worth of publication but some revision is needed before acceptation.

It is mandatory to report the ethical approval details in the manuscript.

The authors mention calf “health” and “behavior”, actually, health measurements was almost only based on BRD morbidity and weight gain and, similarly, the only observed behavior was the lying\standing one. It should have been interesting to determine some biochemical parameters and to make some behavioral  test to assess the effects of the rearing system. If such data are not available, I suggest to change the title and some subtitles (see below) to better define the trial design.

Also, details about calves feeding at birth should be added. Colostrum IgG content is critical in determining weight gain and health status in ruminants, if no data are available to check a relation between colostrum and performance, this concern should be mentioned in the discussion at least as a future aim. I suggest two papers to improve this aspect:

Effect of passive transfer status on preweaning growth performance in dairy goat kids. Massimini G. et al. Journal of the American Veterinary Medical Association, 2007, 231(12).

Effects of passive transfer status on growth performance in buffalo calves. Mastellone, V. et al. Asian-Australasian Journal of Animal Sciences, 2011, 24(7).

Lines 2-3 – add lying\standing behavior, change health in morbidity

Line 108 – please define which colostrum (mother’s, frozen?) and how was it administered to calves. If available add data concerning colostrum IgG content.

Line 148 – lying and standing behavior

Line 161 – Move here line 170-171, clearly indicating (all together) blood collection details (days, time, fasting etc.)

Line 170 – see above

Line 219 – Change health status with morbidity

Line 270 – change the title as suggested above

Author Response

  1. It is mandatory to report the ethical approval details in the manuscript.

We added the details in 2.1, “All experimental produces were approved by Institutional Animal Care and Committee (LACUC Approval No:AW90901202-5-1) of China Agricultural University.”

  1. The authors mention calf “health” and “behavior”, actually, health measurements was almost only based on BRD morbidity and weight gain and, similarly, the only observed behavior was the lying\standing one. It should have been interesting to determine some biochemical parameters and to make some behavioral  test to assess the effects of the rearing system. If such data are not available, I suggest to change the title and some subtitles (see below) to better define the trial design.

We changed the title and some subtitles according the comments.  

  1. Also, details about calves feeding at birth should be added. Colostrum IgG content is critical in determining weight gain and health status in ruminants, if no data are available to check a relation between colostrum and performance, this concern should be mentioned in the discussion at least as a future aim. I suggest two papers to improve this aspect:

We did not detect the colostrum IgG. Thus, no data are available to check a relation between colostrum and performance. We added the related content in the discussion.

  1. Lines 2-3 – add lying\standing behavior, change health in morbidity

Changed accordingly.

  1. Line 108 – please define which colostrum (mother’s, frozen?) and how was it administered to calves. If available add data concerning colostrum IgG content.

The colostrum was frozen, and colostrum was maually bottle-fed to calves. We did not detect the colostrum IgG.

  1. Line 148 – lying and standing behavior

Changed accordingly.

  1. Line 161 – Move here line 170-171, clearly indicating (all together) blood collection details (days, time, fasting etc.)

 We detele the sentence—“Blood samples were collected during the trail”. The blood were collected at at 7, 21, and 42 days of age, and from 8:00 am to 10:00 pm.

  1. Line 170 – see above

 Changed accordingly.

  1. Line 219 – Change health status with morbidity

 Changed accordingly.  

  1. Line 270 – change the title as suggested above

 Changed accordingly.

Reviewer 2 Report

Although the manuscript possesses a significant experimental design, and critical findings, the manuscript needs some revisions as the following:

In the Title, I suggest to change the title into an informative title and to add the most common climate condition in study zone. 

Write the aim of the study clearly in the Abstract.

In the Introduction, write the cited information in a present tense and throughout the manuscript.

In the Methodology and Result Section, the authors should mention the wind speed and calculate the Wind Chell Index as they performed the experimental work during winter season.

Give a description for the claf jackets (Brand, etc.)

The authors should use the past tense in the Material and Result Sections as they can.

Line 161-165: Use the active voice.

Line 266: correct "×g into "xg" 

The authors should remove the repeated numbers in the text that already were written (mentioned) in Table 3. If necessary, put them in the text as percentage compared to the other experimental groups.  

Write a footnote for Table 4.

What is the difference between data Table 4 and Figure 5?

Change the title of Figure 5 into: "Concentration of biochemical parameters of calves in different housing systems." 

 Then write the identification of  A, B, C, etc. under footnote.

In the Discussion:

a. Convert the sub-headings into introductory phrases.

b. Start the Discussion with the main message (answers for your aims that you mentioned in the Introduction)  

Reference no. 24: write the title in a Sentence case font

Reference no. 28: write the Journal Abbreviation in a correct form

Reference no. 30: delete [J] (line 410)

Reference no. 35: write the Journal Abbreviation in a correct form

Reference no. 37: write the Journal Name in a correct case.

based on the previous comments, I recommend to reconsider the manuscript for publication after minor revisions.

Author Response

  1. In the Title, I suggest to change the title into an informative title and to add the most common climate condition in study zone. 

We changed the title to “Effect of rearing systems on growth performance, lying/ standing behaviour, morbidity, and immunity parameters of pre-weaned dairy calves in continental zone in winter”

  1. Write the aim of the study clearly in the Abstract.

We added the aim of the study in the abstract, “To evaluated the effect of all three of these commonly used rearing practices on calves, the experiment was conducted.”

  1. In the Introduction, write the cited information in a present tense and throughout the manuscript.

We checked the tense of the cited information in the full text and revised it according to the comment.

  1. In the Methodology and Result Section, the authors should mention the wind speed and calculate the Wind Chell Index as they performed the experimental work during winter

In the experiment. The wind speed was not measured, so the Wind Chell Index could not be calculated.

  1. Give a description for the claf jackets (Brand, etc.)

The calf jacktes are home-made in farm.

  1. The authors should use the past tense in the Material and Result Sections as they can.

We checked the tense of the cited information in the full text and revised it according to the comment.

  1. Line 161-165: Use the active voice.

 Changed accordingly.

  1. Line 266: correct "×g into "" .The authors should remove the repeated numbers in the text that already were written (mentioned) in Table 3. If necessary, put them in the text as percentage compared to the other experimental groups.  

 Changed accordingly.

  1. Write a footnote for Table 4.

We added a footnote.

  1. What is the difference between data Table 4 and Figure 5?

Compared to Table 4, Figure 5 adds the significance analysis between aeach sampling for each immune index in calves.

  1. Change the title of Figure 5 into: "Concentration of biochemical parameters of calves in different housing systems."  Then write the identification of  A, B, C, etc. under footnote.

 Changed accordingly.

In the Discussion:

  1. Convert the sub-headings into introductory phrases.

We changed “4.1 Ambient temperature and relative humidity” into “4.1 Ambient temperature and relative humidity in dairy farm”; and changed “4.2 Health status, growth performance, lying and standing behavior, and immunity parameters” to “4.2 Health status, growth performance, lying and standing behavior, and immunity parameters of tested calves”

  1. Start the Discussion with the main message (answers for your aims that you mentioned in the Introduction)  

We added the main message before 4.1. “This study systematically evaluated all three of the housing options (outdoor hutch housing, indoor group housing, and indoor individual housing)that are currently used to raise pre-weaned dairy calves in winter. “

  1. Reference no. 24: write the title in a Sentence case font

 Changed accordingly.

  1. Reference no. 28: write the Journal Abbreviation in a correct form

 Changed accordingly.

  1. Reference no. 30: delete [J] (line 410)

 Changed accordingly.

  1. Reference no. 35: write the Journal Abbreviation in a correct form

 Changed accordingly.

  1. Reference no. 37: write the Journal Name in a correct case.

 Changed accordingly.